**EMBO**
Molecular Medicine

# Kaposiform lymphangiomatosis effectively treated with MEK inhibition

Jessica B Foster[1],* , Dong Li[2], Michael E March[2], Sarah E Sheppard[2], Denise M Adams[4], Hakon Hakonarson[2,3] & Yoav Dori[5],**

## Abstract

**Kaposiform lymphangiomatosis (KLA) is a rare lymphatic anomaly primarily affecting the mediastinum with high mortality rate. We present a patient with KLA and significant disease burden harboring a somatic point mutation in the *Casitas B lineage lymphoma* (*CBL*) gene. She was treated with MEK inhibition with complete resolution of symptoms, near-complete resolution of lymphatic fluid burden, and remodeling of her lymphatic system. While patients with KLA have been reported to harbor mutations in *NRAS*, here we report for the first time a causative mutation in the *CBL* gene in a patient with KLA, successfully treated with Ras pathway inhibition.**

**Keywords** CBL proto-oncogene; kaposiform lymphangiomatosis; lymphatic abnormality; MAP kinase signaling system; trametinib
**Subject Categories** Vascular Biology & Angiogenesis

See also: **MT Dellinger & FX McCormack** (October 2020)

## Introduction

Kaposiform lymphangiomatosis (KLA) is a rare disorder that results in abnormal lymphatic vasculature, most commonly in the chest and mediastinum. KLA is typically diagnosed in childhood and results in significant risk of respiratory compromise, infections, coagulopathy, and early death (Croteau *et al*, 2014). A recent case series reporting whole-exome sequencing in 11 patients identified a somatic *NRAS* p.Q61R variant in the lesional tissue of 10 out of 11 cases (Barclay *et al*, 2019). In addition, recently, Li *et al* (2019) reported on MEK inhibition in a patient with *ARAF* mutation and lymphatic conduction disorder leading to near-complete remodeling of the patient's lymphatic system and resolution of symptoms. While this work has prompted investigation into Ras pathway targeting for KLA, no patients to date have been treated and no clinical trials exist. We present an 18-year-old female patient with refractory KLA who did not have a somatic mutation in *NRAS* but rather lesional cells revealed a point mutation in the *CBL* gene. She was treated successfully with MEK inhibition, suggesting a novel treatment strategy for this patient population.

## Results

### Baseline patient characteristics

Our patient's symptoms began at age 6, when she was noted to have intermittent dyspnea with activity and frequent infections including sinusitis, bronchitis, and pneumonias. At the age of 10, she developed streptococcal pharyngitis that progressed to septic shock requiring intensive care. At that time, she was found to have pericardial and pleural chylous effusions. She also developed a large blue lump at her right areola. She eventually was referred to a large children's hospital where she underwent biopsy of her lungs and revealed the diagnosis of KLA. Histology showed lymphatic malformation with foci of spindled lymphatic endothelial cells reactive to CD31 and D2-40. MRI showed the extent of her disease included mediastinum, lungs, right breast, and axilla, as well as spleen, and bony involvement in the sacrum and spine. She was initially treated with steroids, vincristine, interferon, and rapamycin. She remained on rapamycin for 8 years with generally stable disease but intermittent hospitalizations due to worsening pleural effusions requiring drainage. At age 18, she elected to stop therapy with rapamycin and subsequently had severe disease recrudescence. Despite restarting rapamycin, prednisone, and vincristine, her pulmonary status remained extremely poor. Functionally, she had severe dyspnea and orthopnea with chronic coughing, and inability to engage in any exercise or exertion beyond activities of daily living. She continued with intermittent large bilateral pleural effusions and intralobular septal thickening on chest CT (Fig 1A). She subsequently underwent T2-weighted lymphatic mapping, which demonstrated bilateral pleural effusions and mediastinal and pulmonary interstitial edema

1 Division of Oncology, Children's Hospital of Philadelphia, Philadelphia, PA, USA
2 Center for Applied Genomics, Children's Hospital of Philadelphia, Philadelphia, PA, USA
3 Divisions of Human Genetics and Pulmonary Medicine, Children's Hospital of Philadelphia, Philadelphia, PA, USA
4 Vascular Anomalies Center, Boston Children's Hospital, Boston, MA, USA
5 Center for Lymphatic Imaging and Interventions, Children's Hospital of Philadelphia, Philadelphia, PA, USA
*Corresponding author. Tel: +1 267 425 1918; E-mail: fosterjb@email.chop.edu
**Corresponding author. Tel: +1 215 590 1790; E-mail: doriy@email.chop.edu

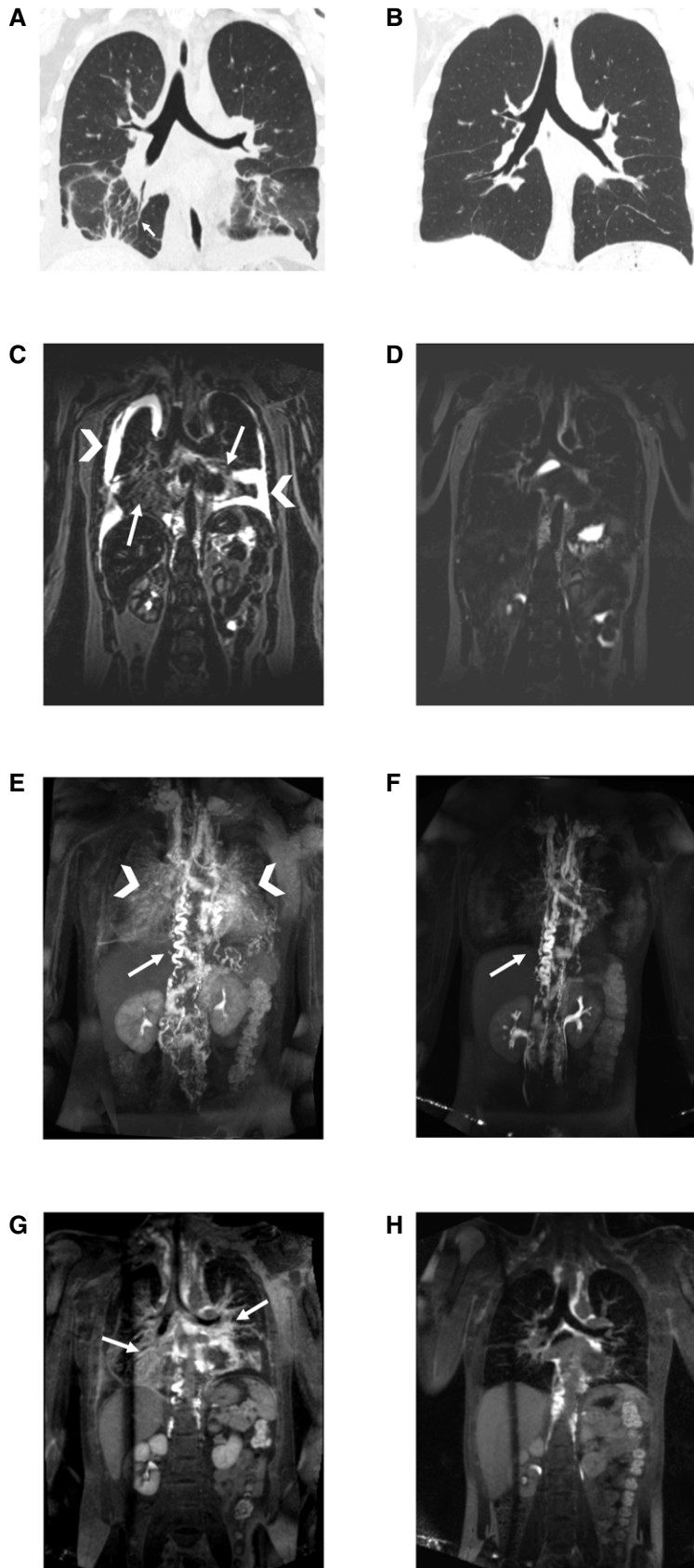

**Figure 1.**

**Figure 1. Imaging before and after treatment with trametinib.**

A, B Coronal CT image of the chest at baseline with pleural fluid and intralobular septal thickening (arrow) (A), and 3 months after starting trametinib (B).

C, D MR T2-weighted coronal slice of the chest at baseline demonstrating bilateral pleural effusions (arrow heads) and pulmonary interstitial edema (arrows) (C), and 3 months on trametinib therapy (D).

E–H IN-DCMRL coronal MIP of the chest at baseline demonstrating dilated and tortuous TD (arrow) with dilated central lymphatic networks and extensive bilateral pulmonary perfusion (arrowheads) (E), and 6 months after trametinib therapy demonstrating dilated and tortuous TD (arrow) with reduction in the extent of the dilated central lymphatic networks and resolution of bilateral pulmonary perfusion (F). IN-DCMRL coronal slice of the chest at the level of the carina demonstrating extensive mediastinal and pulmonary interstitial perfusion (arrows) at baseline (G), and after 6 months of treatment with trametinib (H).

(Fig 1C), and intranodal dynamic contrast lymphangiography (IN-DCMRL) demonstrated massively dilated and tortuous thoracic duct with severe bilateral pulmonary interstitial perfusion and mediastinal perfusion (Fig 1E and G). Pulmonary function tests (PFTs) revealed severe restrictive lung disease (Table 1). Her D-dimer was also elevated as previously reported with KLA patients (Table 1), and her platelets intermittently dropped with disease exacerbations although her other coagulation parameters were normal.

## MEK inhibition repairs respiratory functional status and restructures abnormal lymphatic vessels

Based on the patient's poor quality of life and concern for imminent deterioration, we obtained expanded access to trametinib (*Mekinist*, Novartis) to target the Ras pathway. She was started on a low dose of 0.5 mg daily (0.01 mg/kg/dose) for tolerability for the first 28-day cycle, with plans to escalate to the phase 2 pediatric dosage of 0.025 mg/kg/dose daily with subsequent cycles. However, within 4 weeks of starting the medication the patient reported she subjectively felt much improved with resolution of coughing, dyspnea, and orthopnea. Her exercise tolerance markedly improved, and she started training for a 5K race during the second cycle of therapy. Based on her subjective improvement, she was maintained on the low dose of 0.5 mg daily for three cycles. Follow-up PFTs after cycle 3 revealed significant improvement in her restrictive lung disease as well as normalization of her D-dimer (Table 1). Chest CT revealed near-complete resolution of her pleural effusions and intralobular septal thickening (Fig 1B). After six cycles, she had further improvement in her restrictive lung disease based on PFTs (Table 1), and repeat T2 lymphatic mapping revealed resolution of pleural effusions and pulmonary interstitial edema (Fig 1D) and IN-DCMRL demonstrated resolution of pulmonary interstitial perfusion and marked improvement in mediastinal perfusion (Fig 1F and 1H). She continued to show a dilated and tortuous thoracic duct, but significantly reduced mediastinal collateral flow. Her prior requirement for intermittent drainage of chylous effusions completely resolved. Overall, therapy was well-tolerated with only grade 1 adverse events including acneiform rash, brief eosinophilia that quickly resolved, and brief hematuria and proteinuria that also quickly resolved. She continues on 0.5 mg trametinib daily currently, now 12 months after starting therapy and continued resolution of symptoms.

## Identification of CBL mutation

Prior to initiation of trametinib therapy, the patient had re-accumulation of pleural chylous effusions, which were drained and fluid collected for research. Deep coverage WES of >330× detected a novel somatic mosaic variant, c.2322T>G:p.Y774*, at 4% frequency, in the *CBL* gene (Fig 2A), which was subsequently shown by droplet digital

**Table 1. Laboratory and pulmonary function tests before and after treatment with MEK inhibition.**

| | Baseline | 3 months after treatment | 6 months after treatment |
|---|---|---|---|
| White blood cells (K/µl) | 13.7 | 7.2 | 10 |
| Hemoglobin (g/dl) | 16 | 13.9 | 13.7 |
| Platelets (K/µl) | 150 | 243 | 231 |
| D-dimer (µg/ml FEU) | 9.17 | 0.49 | < 0.27 |
| FVC (l) (% predicted) | 1.82 48 | 2.71 71 | 3.03 81 |
| FEV1 (l) (% predicted) | 1.75 52 | 2.65 79 | 2.98 90 |
| FEV1/FVC (l) (% predicted) | 96 110 | 98 113 | 98 110 |
| FEF 25–75 (l) (% predicted) | 5.05 133 | 6.49 172 | 6.85 174 |
| TLC (l) (% predicted) | 3.02 67 | 4.04 80 | 4.36 87 |
| FRC (l) (% predicted) | 2.01 84 | 2.66 97 | 2.95 109 |
| RV (l) (% predicted) | 1.10 90 | 1.33 113 | 1.33 115 |
| RV/TLC (%) (% predicted) | 36 133 | 33 150 | 31 141 |
| DLCO (ml/min/mmHg) (% predicted) | 12.57 54 | 19.30 74 | 20.41 94 |

DLCO, diffusing capacity of the lungs for carbon monoxide; FEF 25–75, forced expiratory flow at 25–75% forced vital capacity; FEV1, forced expiratory volume in 1 s; FRC, functional residual capacity; FVC, forced vital capacity; RV, residual volume; TLC, total lung capacity.

polymerase chain reaction (ddPCR) to be present at a low level (Fig 2B). The DNA sample from the CD31 cells was exhausted during the experimental process, and the input sample for the ddPCR experiment was suboptimal and could not be repeated. As the mutation is predicted to truncate the protein, which is a negative regulator of Ras signaling thus in turn exerting gain-of-function effects in the Ras-MAPK signaling pathway, and the patient demonstrated robust response to therapy, we elected not to pursue a new sample. All other biological samples (blood, saliva, skin fibroblasts, normal lymph node of the groin) were mutation-negative.

## Discussion

Kaposiform lymphangiomatosis is a devastating disease, marked by characteristic intrathoracic lymphatic anomalies and hemorrhagic-

**A**

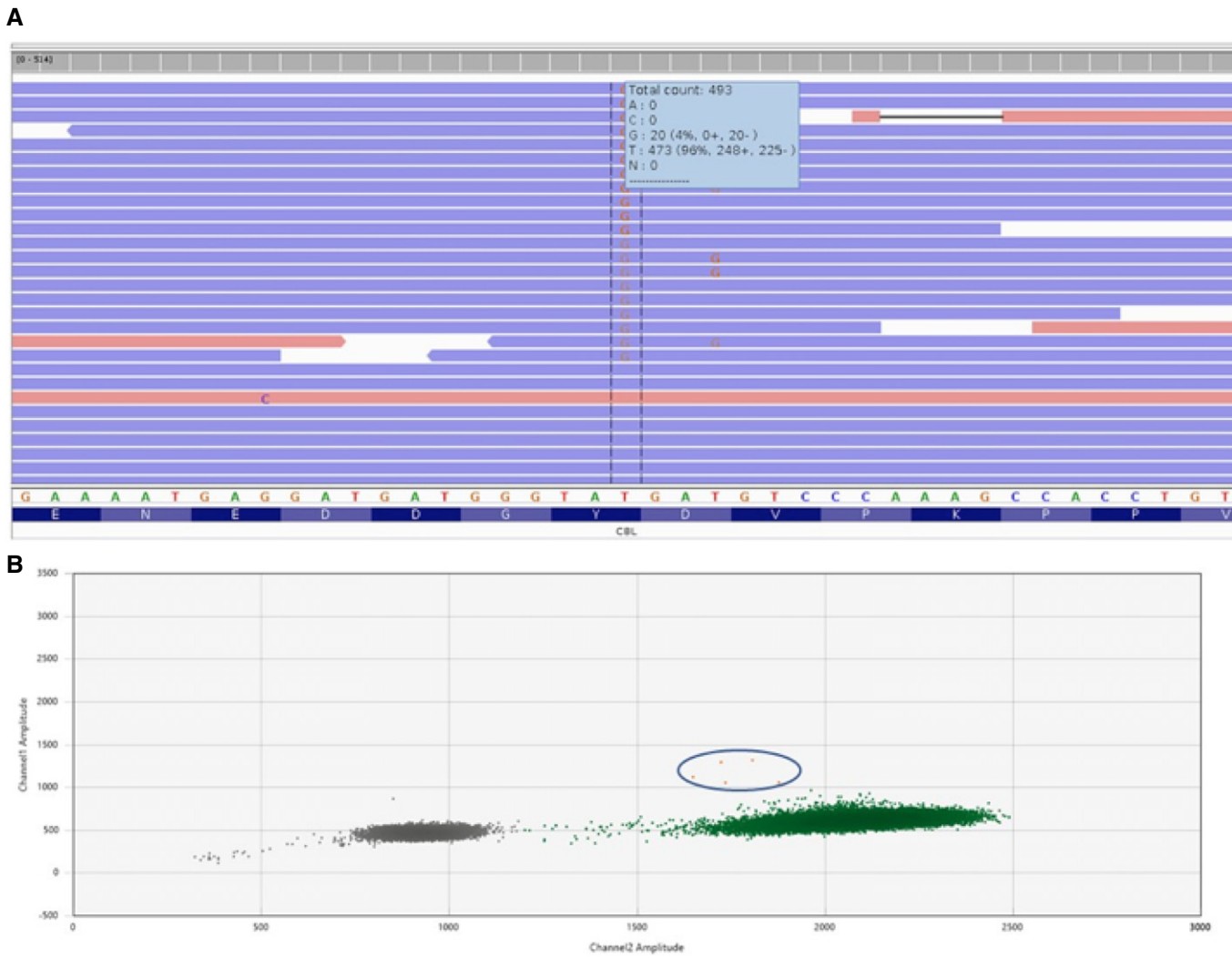

**Figure 2. Mutation analysis for the KLA patient uncovered a novel variant in the CBL gene.**

A   Deep coverage exome sequencing Integrative Genomics Viewer (IGV) view of CBL c.2322T>G in DNA sample that was derived from CD31-selected cells from pleural effusion fluid. The alternate allele fraction was 4% (20/493).

B   Droplet digital polymerase chain reaction (ddPCR) result with five mutant positive droplets (circle) from a low concentration leftover sample which resulted in ~ 0.005% fractional abundance.

chylous effusions, with spindled lymphatic endothelial cells on histology (Croteau *et al*, 2014). The International Society for the Study of Vascular Anomalies (ISSVA) classifies KLA as a subset within complex lymphatic malformations. The vast majority of diagnoses are made in children, but rare reports of KLA have been reported in adults as well with a similar presentation (Safi *et al*, 2014). KLA has an overall survival of 34% despite aggressive multimodal therapy (Croteau *et al*, 2014). Many of these patients have been treated with rapamycin after success was reported generally for complex vascular anomalies, including KLA (Hammill *et al*, 2011; Adams *et al*, 2016). In addition, lymphatic cells isolated from patients with KLA were shown to have gain-of-function mutations in genes involved in the PI3K-AKT-mTOR pathway with decreased cell growth in response to inhibitors of these pathways (Boscolo *et al*, 2019). Genomic evaluation of vascular anomalies such as KLA

has only just begun in recent years, with the first case series identifying *NRAS* p.Q61R as the likely driver for the majority of patients (Barclay *et al*, 2019). Similarly, other groups have identified *NRAS* p.Q61R from cell-free DNA in biofluids of affected patients (Ozeki *et al*, 2019). In this report, we describe a patient who did not harbor the *NRAS* p.Q61R mutation, but rather a mutation upstream in the Ras pathway, *CBL* c.2322T>G:p.Y774*. We suspect this mutation is the driver for her lymphatic anomaly, and it is unlikely to have been induced by her prior chemotherapy, which included vincristine, prednisone, and rapamycin. *CBL* is a proto-oncogene with mutations identified in juvenile myelomonocytic leukemia and acute lymphoblastic leukemia, as well as Noonan syndrome. CBL proteins target tyrosine receptor kinases for degradation, and loss-of-function mutations are associated with stabilization of the kinases and subsequent activation of the Ras pathway.

## The paper explained

### Problem
Kaposiform lymphangiomatosis (KLA) is a rare lymphatic anomaly affecting the lungs and mediastinum of patients with a high mortality rate. No curative therapy exists.

### Results
We identified a patient with a lesional mutation in CBL who was effectively treated with MEK inhibition.

### Impact
Mutations in the Ras pathway, including CBL, are likely driving KLA and potentially offer a targeted therapeutic strategy for this disease.

The patient in this study experienced near-complete resolution of symptoms and extensive remodeling of her lymphatic system within months of initiation of therapy, including resolution of prior chronic chylous effusion. The change in her clinical status was evident in her follow-up imaging and pulmonary function tests. Of note, her improved pleural effusions from trametinib therapy may have also contributed to her improved PFT results. The resolution of lung disease with lymphatic remodeling is remarkable and potentially should change how we evaluate and treat lung disease in this patient population. The relatively low dose required for dramatic improvement in this case highlights that a small amount may be sufficient to treat lymphatic disorders, which will likely limit the side effect profile compared with the higher doses used for oncologic cases. Ongoing prospective studies are now in preparation to evaluate Ras pathway inhibition in clinical trials of large cohorts of patients. In addition, these results offer hope to other patients with lymphatic-induced lung disease and warrant further investigation.

## Materials and Methods

All studies were completed on an IRB-approved protocol, and the patient gave informed consent. All experiments conformed to the principles set out in the WMA Declaration of Helsinki and the Department of Health and Human Services Belmont Report. Patient endothelial cells were selected by cell sorting of CD31-positive cells from her pleural fluid and expanded and grown in culture. While not cell-specific, CD31 was used for selection as it captures all endothelial cells, and in particular captures the abnormal spindle cells in KLA. We conducted exome capture on DNA isolated from cells after expansion in culture using the Twist Human Core Exome Kit (TWIST Bioscience), guided by the manufacturer's protocols. The libraries were subsequently sequenced for 101 cycles with a paired-end mode on the Illumina NovaSeq 6000. All the raw reads were aligned to the reference human genome using the Burrows–Wheeler Aligner (BWA), and single-nucleotide variants and small insertions/deletions were detected using the MuTect2, a somatic variant caller in GATK package that uses local reassembly of haplotypes method to detect low-frequency somatic mutations. The mean coverage depth was 338-fold. ANNOVAR and SnpEff were used to functionally annotate the variants. Subsequent gene prioritization was performed on the basis of deleterious prediction and biological relevance by referring to the Online Mendelian Inheritance in Man

database, which resulted in a somatic *CBL* nonsense mutation as the most likely disease-causing candidate. ddPCR was performed on DNA with somatic candidate mutation and DNA isolated from peripheral blood mononuclear cells using the mutation detection assay: *CBL* pY774*, Homo sapiens (Assay ID: dHsaMDS412920293, Bio-Rad). Each DNA sample was mixed with a primer pair and two TaqMan probes, FAM (mutant sequence), and HEX (wild-type sequence) per manufacturer's instruction. Droplets were made using the Bio-Rad QX200 Droplet Generator (Bio-Rad), and PCR was performed on Bio-Rad QX200 PCR system (Bio-Rad).

## Data availability

The datasets and computer code produced in this study are available in the following database:

- Exome sequencing: dbGAP at NHGRI. Accession Number: phs001802.v2.p1 (https://www.ncbi.nlm.nih.gov/projects/gap/cgi-bin/study.cgi?study_id=phs001802.v2.p1).

**Expanded View** for this article is available online.

## Acknowledgements
This work was supported by Institute Development Fund to the Center for Applied Genomics from the Children's Hospital of Philadelphia. Research reported in this publication was supported by the National Center for Advancing Translational Sciences of the National Institutes of Health under award number TL1TR001880. The content is solely the responsibility of the authors and does not necessarily represent the official views of the National Institutes of Health. Research reported in the publication was supported by the Institute for Translational Medicine and Therapeutics of the Perelman School of Medicine at the University of Pennsylvania.

## Author contributions
JBF, DMA, and YD designed the study. JBF, DL, MEM, SES, DMA, HH, and YD acquired and interpreted the data. JBF, DMA, and YD treated the patient. JBF drafted the manuscript, and all authors revised and approved the manuscript.

## Conflict of interest
H.H. is a scientific advisor to Aevi Genomic Medicine Inc., and he and CHOP own shares in the company. The other authors declare no competing interests.

## For more information
https://www.lgdalliance.org/

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
