## [Review Process File · EMBO Molecular Medicine]

Kaposiform lymphangiomatosis effectively treated with MEK inhibition

Jessica Foster, Dong Li, Michael March, Sarah Sheppard, Denise Adams, Hakon Hakonarson, and Yoav Dori

DOI: [10.15252/emmm.202012324](https://doi.org/10.15252/emmm.202012324)

Corresponding authors: Jessica Foster (fosterjb@email.chop.edu) , Yoav Dori (doriy@email.chop.edu)

Review Timeline:

Submission Date:	24th Mar 20
Editorial Decision:	23rd Apr 20
Revision Received:	14th May 20
Editorial Decision:	19th May 20
Revision Received:	19th Jul 20
Accepted:	31st Jul 20

Editor: Zeljko Durdevic

Transaction Report:

Thank you for the submission of your manuscript to EMBO Molecular Medicine. We have now heard back from the three referees who agreed to evaluate your manuscript. As you will see from the reports below, the referees are positive and find the study interesting and important. However, they also have a few suggestions and some minor criticisms that I would like you to address in a revision of the current manuscript. Particular attention should be given to providing the rationale for using CD31+ cells for genetic analysis and to discussing the future approaches to large patient cohorts' trials.

Addressing the reviewers' concerns in full will be necessary for further considering the manuscript in our journal. Please note that EMBO Molecular Medicine encourages a single round of revision only and therefore, acceptance or rejection of the manuscript will depend on the completeness of your responses included in the next, final version of the manuscript. For this reason, and to save you from any frustrations in the end, I would strongly advise against returning an incomplete revision.

We would welcome the submission of a revised version within three months for further consideration. However, we realize that the current situation is exceptional on the account of the COVID-19/SARS-CoV-2 pandemic. Please let us know if you require longer to complete the revision.

I look forward to receiving your revised manuscript.

***** Reviewer's comments *****

Referee #1 (Remarks for Author):

Thank you for the opportunity to review this manuscript. The authors showed the significant improvements of the symptoms KLA patient treated with MEK inhibitor. This manuscript presents an interesting case report about novel treatment for KLA and other lymphatic anomalies having RAS gene abnormalities.

1. I wonder how the authors diagnose KLA, not other lymphatic diseases? What is the definition? The diagnoses of KLA are based on clinical presentation/features, imaging studies and/or histological data.

2. WES detected a low frequent somatic mutation in CBL gene. The mutated DNA was isolated from CD31 cells of pleural chylous effusions. Why did you use these cells? The author should indicate the association these mutated cells and the pathogenesis of the KLA patient. And, do you have samples from the thoracic or other affected lesions?

3. How do you consider the possibility that the CBL gene mutation might be associated with chemotherapy? Although there's a very low probability of this problem occurring.

I hope that my comment is very useful for the improvement of the article.

Referee #2 (Remarks for Author):

It was a great pleasure reading this very interesting report. Lymphangiomas are an ultrarare medical condition existing as Kaposiform "variant" in children but also in adulthood.

As therapeutic options are very limited, the findings reported here may offer new therapeutic options to some patients with (Kaposiform) lymphangiomas.

Main limitation of this casuistic lies in the nature of a report on a single patient and authors should at least discuss approaches for larger future patient cohorts. Moreover, authors are asked to include a short statement on the "adult" form of lymphangiomas.

the quality of the HRCT showing the interlobular septal thickening is limited and authors may try to provide a better resolution and especially a slice where the interlobular septa can be seen more pronounced.

the reporting of the lung functional parameters is quite unusual. Authors should report absolute values (in l) and %predicted and discuss that improvements seen might also be due to the decrease of pleural effusion

Referee #3 (Remarks for Author):

This is a case report of a patient with KLA who was found to have a CBL mutation and who responded in a dramatic fashion with lymphatic remodeling and resolution of chylos complication to MEK inhibition. Similar responses have been reported in KLA patients with RAS mutations treated with mTOR inhibitors and ARAF mutations treated with MEK inhibition. The novelty with this case report is that the putative causative mutation was in an effector that is upstream of RAS and known to be associated with Noonan's syndrome and leukemias. The report is well written. I have a few comments:

As acknowledged, DNA analysis performed on an expanded endothelial cell population (CD31++) from chylos fluid was limited to some extent because of low allele frequency and scarcity of sample. The pulmonary function test results published as z-scores will be unfamiliar to most clinicians--would favor a more conventional approach (such as FEV1 in liters or percent predicted). It would be interesting to know if the non-CD31+ cells in the chylos fluid were mutation positive or negative. It should be mentioned whether the patient's requirement for intermittent drainage of chylos fluids completely resolved. It bears emphasizing the low dose therapy with MEK inhibitors may be sufficient, limiting the side effects that are known to plague treatment of melanoma and other neoplasms

Referee #1 (Remarks for Author):

Thank you for the opportunity to review this manuscript. The authors showed the significant improvements of the symptoms KLA patient treated with MEK inhibitor. This manuscript presents an interesting case report about novel treatment for KLA and other lymphatic anomalies having RAS gene abnormalities.

1. I wonder how the authors diagnose KLA, not other lymphatic diseases? What is the definition? The diagnoses of KLA are based on clinical presentation/features, imaging studies and/or histological data.

Thank you for this comment. In this case, the diagnosis was made by histology after biopsy, which is the gold standard. The diagnosis was initially made at an outside institution, and the slides were sent to our institution where we also confirmed the diagnosis of KLA based on histologic appearance with CD31 and D2-40 reactivity in the foci of spindled cells. This is updated in the manuscript on page 4 to be more clear. It should be noted the diagnosis was suspected based on her symptoms of persistent pneumonia with bloody pleural effusions and mild coagulopathy. In our experience many cases of KLA are diagnosed by clinical history and radiologic appearance, with biopsy if safe and feasible to do so. Many patients have a mild coagulopathy with low platelet count and fibrinogen, which can make biopsy challenging. The original case series that defined KLA (see reference Croteau et al 2014) reported the characteristic clinical findings and the defining histologic appearance. We have updated the discussion on page 6 to reflect this.

2. WES detected a low frequent somatic mutation in CBL gene. The mutated DNA was isolated from CD31 cells of pleural chylous effusions. Why did you use these cells? The author should indicate the association these mutated cells and the pathogenesis of the KLA patient. And, do you have samples from the thoracic or other affected lesions?

Thank you for pointing this out and we have clarified in the text on page 8. CD31 cell sorting was chosen to select for all endothelial cells, which would include lymphatic cells including abnormal KLA cells. No further selection was completed, which is likely why the mutant allele frequency was so low, as there was a significant number of normal endothelial cells and potentially other cells such as monocytes that also stain positive for CD31. In our experience

this is frequently the case for somatic mutations in vascular anomalies. We attempted to extract DNA from the primary biopsy sample, but as the sample was obtained in 2012 it was significantly degraded and did not pass quality checks. All other samples tested including saliva, blood, normal lymph node biopsy and skin biopsy were negative for the mutation as noted on page 5.

3. How do you consider the possibility that the CBL gene mutation might be associated with chemotherapy? Although there's a very low probability of this problem occurring.

Thank you for this comment as well. We agree it is incredibly low likelihood that chemotherapy induced the CBL mutation. The chemotherapy she received included vincristine, prednisone, and rapamycin, none of which are associated with inducing point mutations. We have updated the discussion to include this point on page 7.

I hope that my comment is very useful for the improvement of the article.

Referee #2 (Remarks for Author):

It was a great pleasure reading this very interesting report. Lymphangiomatosis is an ultrarare medical condition existing as Kaposiform "variant" in children but also in adulthood.

As therapeutic options are very limited, the findings reported here may offer new therapeutic options to some patients with (Kaposiform) lymphangiomatosis.

Main limitation of this casuistic lies in the nature of a report on a single patient and authors should at least discuss approaches for larger future patient cohorts.

Thank you so much for this comment, and we have added to our discussion larger clinical trials which are now starting based on this work (page 8).

Moreover, authors are asked to include a short statement on the "adult" form of lymphangiomatosis.

Thank you for this comment, we have added a description of adults diagnosed with KLA, which is a rare phenomenon but has been described (page 6).

The quality of the HRCT showing the interlobular septal thickening is limited and authors may try to provide a better resolution and especially a slice where the interlobular septa can be seen more pronounced.

Thank you, we have updated the figure.

The reporting of the lung functional parameters is quite unusual. Authors should report absolute values (in l) and %predicted and discuss that improvements seen might also be due to the decrease of pleural effusion

Thank you again, we have updated to the absolute values and percent predicted (page 14). We have also added a comment to discuss that the improvement may also be due to the decrease in pleural effusion (page 7-8).

Referee #3 (Remarks for Author):

This is a case report of a patient with KLA who was found to have a CBL mutation and who responded in a dramatic fashion with lymphatic remodeling and resolution of chylous complication to MEK inhibition. Similar responses have been reported in KLA patients with RAS mutations treated with mTOR inhibitors and ARAF mutations treated with MEK inhibition. The novelty with this case report is that the putative causative mutation was in an effector that is upstream of RAS and known to be associated with Noonan's syndrome and leukemias. The report is well written. I have a few comments:

As acknowledged, DNA analysis performed on an expanded endothelial cell population (CD31++) from chylous fluid was limited to some extent because of low allele frequency and scarcity of sample.

Yes thank you, as noted above to reviewer #1: CD31 selection was chosen to select for all endothelial cells, which would include lymphatic cells including abnormal KLA cells. No further selection was completed, which is likely why the mutant allele frequency was so low, as there was a significant number of normal endothelial cells and potentially other cells such as monocytes that also stain positive for CD31. We attempted to extract DNA from the primary biopsy sample, but as the sample was obtained in 2012 it was significantly degraded and did not pass quality checks.

The pulmonary function test results published as z-scores will be unfamiliar to most clinicians-- would favor a more conventional approach (such as FEV1 in liters or percent predicted).

Thank you for this comment, we have updated the table to absolute values and percent predicted (page 14).

It would be interesting to know if the non-CD31+ cells in the chylous fluid were mutation positive or negative.

Unfortunately we did not save non-CD31+ cells from the chylous fluid; we only selected CD31+ cells for expansion. However we are able to comment that all other types of cells tested including saliva, blood, normal lymph node biopsy and skin biopsy were negative for the mutation as noted on page 5.

It should be mentioned whether the patient's requirement for intermittent drainage of chylous fluids completely resolved.

Yes, thank you for pointing this out, the patient's requirement for intermittent drainage completely resolved and she remains without this requirement to this day. We have updated on page 6.

It bears emphasizing the low dose therapy with MEK inhibitors may be sufficient, limiting the side effects that are known to plague treatment of melanoma and other neoplasms

Thank you again, this is a very important point and we have added this to the discussion on page 8.

19th May 2020

Dear Dr. Foster,

Thank you for the submission of your revised manuscript to EMBO Molecular Medicine. I am pleased to inform you that we will be able to accept your manuscript pending the following final amendments:

The authors performed the requested editorial changes.

31st Jul 2020

Dear Dr. Foster,

We are pleased to inform you that your manuscript is accepted for publication and is now being sent to our publisher to be included in the next available issue of EMBO Molecular Medicine pending the submission of the URL for your WES data. Please send us the URL as soon as you receive it from the NHGRI.

Corresponding Author Name: Jessica Foster

Manuscript Number: EMM-2020-12324